# A diversity of localized timescales in network activity

**Rishidev Chaudhuri[1,2], Alberto Bernacchia[3], Xiao-Jing Wang[2,4]\***

[1]Department of Applied Mathematics, Yale University, New Haven, United States; [2]Department of Neurobiology, Yale University, New Haven, United States; [3]School of Engineering and Science, Jacobs University Bremen, Bremen, Germany; [4]Center for Neural Science, New York University, New York, United States

**Abstract** Neurons show diverse timescales, so that different parts of a network respond with disparate temporal dynamics. Such diversity is observed both when comparing timescales across brain areas and among cells within local populations; the underlying circuit mechanism remains unknown. We examine conditions under which spatially local connectivity can produce such diverse temporal behavior.

In a linear network, timescales are segregated if the eigenvectors of the connectivity matrix are localized to different parts of the network. We develop a framework to predict the shapes of localized eigenvectors. Notably, local connectivity alone is insufficient for separate timescales. However, localization of timescales can be realized by heterogeneity in the connectivity profile, and we demonstrate two classes of network architecture that allow such localization. Our results suggest a framework to relate structural heterogeneity to functional diversity and, beyond neural dynamics, are generally applicable to the relationship between structure and dynamics in biological networks.

**\*For correspondence:** xjwang@nyu.edu

**Competing interests:** The authors declare that no competing interests exist.

## Introduction

A major challenge in the study of neural circuits, and complex networks more generally, is understanding the relationship between network structure and patterns of activity or possible functions this structure can subserve (*Strogatz, 2001*; *Newman, 2003*; *Honey et al., 2010*; *Sporns, 2011*). A number of neural networks show a diversity of time constants, namely different nodes (single neurons or local neural groups) in the network display dynamical activity that changes on different timescales. For instance, in the mammalian brain, long integrative timescales of neurons in the frontal cortex (*Romo et al., 1999*; *Wang, 2001*; *Wang, 2010*) are in striking contrast with rapid transient responses of neurons in a primary sensory area (*Benucci et al., 2009*). Furthermore, even within a local circuit, a diversity of timescales may coexist across a heterogeneous neural population. Notable recent examples include the timescales of reward integration in the macaque cortex (*Bernacchia et al., 2011*), and the decay of neural firing rates in the zebrafish (*Miri et al., 2011*) and macaque oculomotor integrators (*Joshua et al., 2013*). While several models have been proposed, general structural principles that enable a network to show a diversity of timescales are lacking.

Studies of the cortex have revealed that neural connectivity decays rapidly with distance (*Holmgren et al., 2003*; *Markov et al., 2011*; *Perin et al., 2011*; *Levy and Reyes, 2012*; *Markov et al., 2014*; *Ercsey-Ravasz et al., 2013*) as does the magnitude of correlations in neural activity (*Constantinidis and Goldman-Rakic, 2002*; *Smith and Kohn, 2008*; *Komiyama et al., 2010*). This characteristic is apparent on multiple scales: in the cerebral cortex of the macaque monkey, both the number of connections between neurons in a given area and those between neurons across different brain areas decay rapidly with distance (*Markov et al., 2011*, *2014*). Intuitively, local connectivity may suggest that the timescales of network activity are localized, by which we mean that nodes that respond with

**eLife digest** Many biological systems can be thought of as networks in which a large number of elements, called 'nodes', are connected to each other. The brain, for example, is a network of interconnected neurons, and the changing activity patterns of this network underlie our experience of the world around us. Within the brain, different parts can process information at different speeds: sensory areas of the brain respond rapidly to the current environment, while the cognitive areas of the brain, involved in complex thought processes, are able to gather information over longer periods of time. However, it has been largely unknown what properties of a network allow different regions to process information over different timescales, and how variations in structural properties translate into differences in the timescales over which parts of a network can operate.

Now Chaudhuri et al. have addressed these issues using a simple but ubiquitous class of networks called linear networks. The activity of a linear network can be broken down into simpler patterns called eigenvectors that can be combined to predict the responses of the whole network. If these eigenvectors 'map' to different parts of the network, this could explain how distinct regions process information on different timescales.

Chaudhuri et al. developed a mathematical theory to predict what properties would cause such eigenvectors to be separated from each other and applied it to networks with architectures that resemble the wiring of the brain. This revealed that gradients in the connectivity across the network, such that nodes share more properties with neighboring nodes than distant nodes, combined with random differences in the strength of inter-node connections, are general motifs that give rise to such separated activity patterns. Intriguingly, such gradients and randomness are both common features of biological systems.

a certain timescale are contained within a particular region of the network. Such a network would show patterns of activity with different temporal dynamics in disparate regions. Surprisingly, this is not always true and, as we show, additional conditions are required for localized structure to translate into localized temporal dynamics.

We study this structure–function relationship for linear networks of interacting nodes. Linear networks are used to model a variety of physical and biological networks, especially those where inter-node interactions are weighted (*Newman, 2010*). Most dynamical systems can be linearized around a point of interest, and so linear networks generically emerge when studying the response of nonlinear networks to small perturbations (*Strogatz, 1994*; *Newman, 2010*). Moreover, for many neurons the dependence of firing rate on input is approximately threshold-linear over a wide range (*Ahmed et al., 1998*; *Ermentrout, 1998*; *Wang, 1998*; *Chance et al., 2002*), and linear networks are common models for the dynamics of neural circuits (*Dayan and Abbott, 2001*; *Shriki et al., 2003*; *Vogels et al., 2005*; *Rajan and Abbott, 2006*; *Ganguli et al., 2008*; *Ganguli et al., 2008*; *Murphy and Miller, 2009*; *Miri et al., 2011*).

The activity of a linear network is determined by a set of characteristic patterns, called eigenvectors (*Rugh, 1995*). Each eigenvector specifies the relative activation of the various nodes. For example, in one eigenvector the first node could show twice as much activity as the second node and four times as much activity as the third node, and so on. The activity of the network is the weighted sum of contributions from the eigenvectors. The weight (or amplitude) of each eigenvector changes over time with a timescale determined by the eigenvalue corresponding to the eigenvector. The network architecture determines the eigenvectors and eigenvalues, while the input sets the amplitudes with which the various eigenvectors are activated. In *Figure 1*, we illustrate this decomposition in a simple schematic network with three eigenvectors whose amplitudes change on a fast, intermediate and slow timescale respectively.

In general, the eigenvectors are poorly segregated from each other: each node participates significantly in multiple eigenvectors and each eigenvector is spread out across multiple nodes (*Trefethen and Embree, 2005*). Consequently, timescales are not segregated, and a large number of timescales are shared across nodes. Furthermore, if the timescales have largely different values, certain eigenvectors are more persistent than others and dominate the nodes at which they are present. If these slow

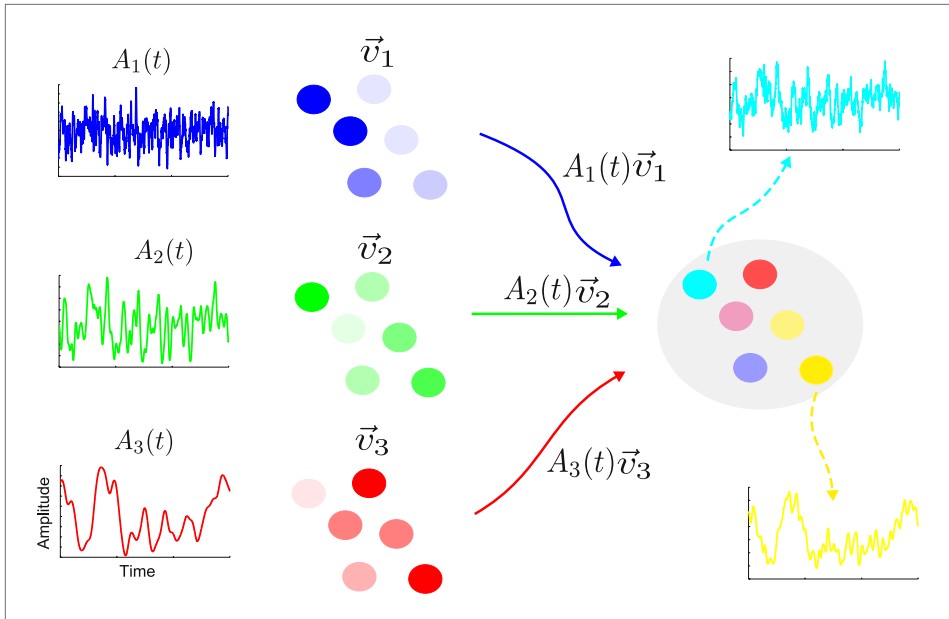

**Figure 1**. The activity of a linear network can be decomposed into contributions from a set of eigenvectors. On the right is shown a sample network along with the activity of two nodes (cyan and yellow). The activity of this network is the combination of a set of eigenvectors whose spatial distributions are shown in blue, green and red on the left. The nodes are colored according to the contributions of the various eigenvectors. Each eigenvector has an amplitude that varies in time with a single timescale given by the corresponding eigenvalue; here the blue, green and red eigenvectors have a fast, intermediate and slow timescale, respectively. The cyan node is primarily a combination of the blue and green eigenvectors; hence its activity is dominated by a combination of the blue and green amplitudes and it shows a fast and an intermediate timescale. Similarly, the yellow node has large components in the green and red eigenvectors, therefore its activity reflects the corresponding amplitudes and intermediate and slow timescales.

timescales are spread across multiple nodes, they dominate the network activity and the nodes will show very similar temporal dynamics. This further limits the diversity of network computation.

In this paper, we begin by observing that rapidly-decaying connectivity by itself is insufficient to give rise to localized eigenvectors. We then examine conditions on the network-coupling matrix that allow localized eigenvectors to emerge and build a framework to calculate their shapes. We illustrate our methods with simple examples of neural dynamics. Our examples are drawn from Neuroscience, but our results should be more broadly applicable for understanding network dynamics and the relationship between the structure and function of complex systems.

## Results

We study linear neural networks endowed with a connection matrix $W$ ($j,k$) ('Methods', *Equation 9*), which denotes the weight of connection from node $k$ to node $j$. For a network with $N$ nodes, the matrix $W$ has $N$ eigenvectors and $N$ corresponding eigenvalues. The time constant associated with the eigenvector $\mathbf{v}_\lambda$ is $1/\mathfrak{Re}(-\lambda)$, where $\lambda$ is the corresponding eigenvalue ('Methods', *Equation 11*). This time constant is present at all nodes where the eigenvector has non-zero magnitude. We say an eigenvector is delocalized if its components are significantly different from 0 for most nodes. In this case, the corresponding timescale is spread across the entire network. On the other hand, if an eigenvector is localized then $\mathbf{v}_\lambda$ ($j$) $\approx 0$ except for a restricted subset of spatially contiguous nodes, and the timescale $1/\mathfrak{Re}(-\lambda)$ is confined to a region of the network. If most or all of the eigenvectors are localized, then different nodes show separated timescales in their dynamical response to external stimulation.

Note that even if the eigenvectors are localized, a large proportion of network nodes could respond to a given input, but they would do so with disparate temporal dynamics. Conversely, even if the eigenvectors are delocalized, a given input could still drive some nodes much more strongly than

others. However, the temporal dynamics of the response will be very similar at the various nodes even if the magnitudes are different.

Consider a network with nodes arranged in a ring, as shown in the top panel of *Figure 2A*. The connection strength between nodes decays with distance according to

$$W(j,k) = e^{-|j-k|/l_c},$$

where, $l_c$ is set to be 1 node so that the connectivity is sharply localized spatially. In *Figure 2B* we plot the absolute values and real parts of three sample eigenvectors. The behavior is typical of all eigenvectors: despite the local connectivity they are maximally delocalized and each node contributes with the same relative weight to each eigenvector (its absolute value is constant, while its real and imaginary parts oscillate across the network). As shown in *Figure 2C*, the timescales of decay are very similar across nodes.

As known from the theory of discrete Fourier transforms, such delocalized eigenvectors are generically seen if the connectivity is translationally invariant, meaning that the connectivity profile is the same around each node (see mathematical appendix [*Supplementary file 1*], Section 1 or standard references on linear algebra or solid-state physics [*Ashcroft and Mermin, 1976*]). In this case the *j*th component of the eigenvector $v_\lambda$ is

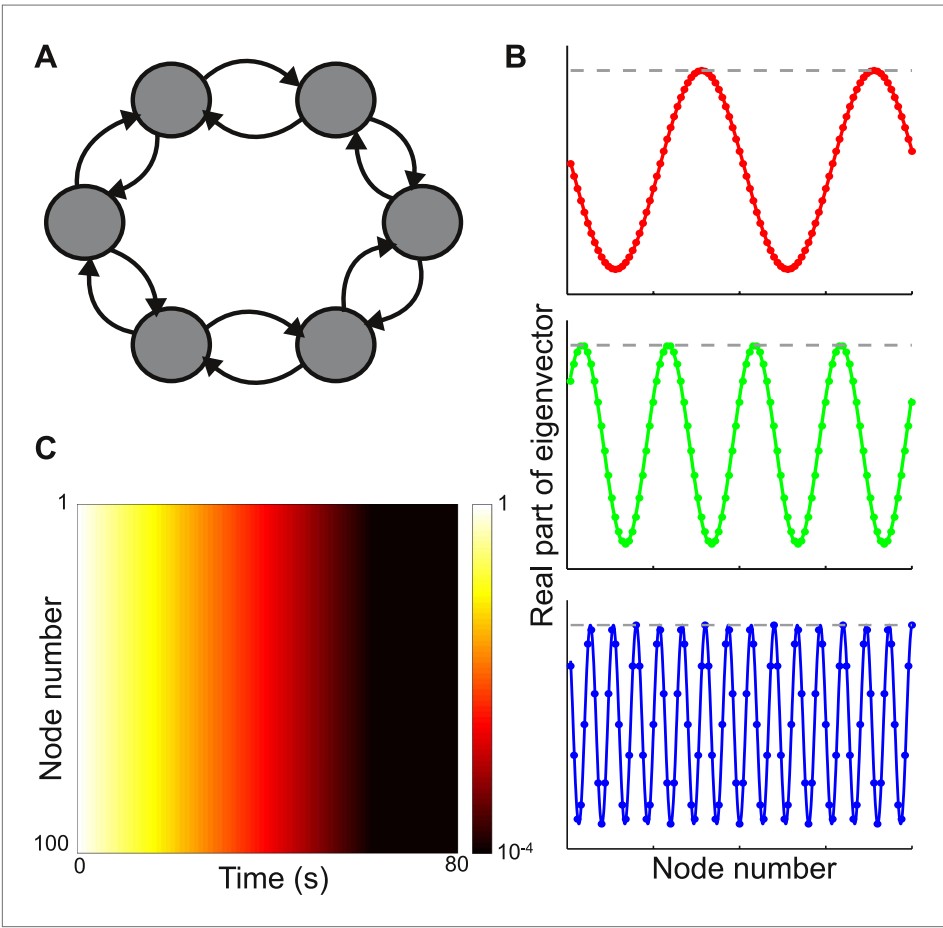

**Figure 2**. Local connectivity is insufficient to yield localized eigenvectors. (**A**) The network consists of 100 nodes, arranged in a ring. Connection strength decays exponentially with distance, with characteristic length of one node, and is sharply localized. The network topology is shown here as a schematic, with six nodes and only nearest-neighbor connections. (**B**) The eigenvectors are maximally delocalized. Three eigenvectors are shown, and the others are similar. The absolute value of each eigenvector, shown with the gray dashed lines, is the same at all nodes. The real part of each eigenvector, shown in color, oscillates with a different frequency for each eigenvector. (**C**) Dynamical response of the network to an input pulse, shown on a logarithmic scale. All nodes show similar response timescales.

$$\mathbf{v}_\lambda(j) = e^{i\omega j}, \tag{1}$$

where, $\omega/2\pi$ is the oscillation frequency (which depends on $\lambda$) and $i$ is the imaginary unit ($i^2 = -1$). Thus local connectivity is insufficient to produce localized eigenvectors.

We developed a theoretical approach that enables us to test network architectures that yield localized eigenvectors. Although in general it is not possible to analytically calculate all timescales (eigenvalues) of a generic matrix, the theory allows us to predict which timescales would be localized and which would be shared. For the localized timescales, it yields a functional form for the shape of the corresponding localized eigenvectors. Finally, the theory shows how changing network parameters promotes or hinders localization. For a further discussion of these issues, see Section 2 of the mathematical appendix (*Supplementary file 1*).

For a given local connectivity, $W(j,k)$, we postulate the existence of an eigenvector $\mathbf{v}_\lambda$ that is well localized around some position, $j_0$, defined as its center. We then solve for the detailed shape (functional form) of our putative eigenvector and test whether this shape is consistent with our prior assumption on $\mathbf{v}_\lambda$. If so, this is a valid solution for a localized eigenvector.

Specifically, if $\mathbf{v}_\lambda$ is localized around $j_0$ then $\mathbf{v}_\lambda(k)$ is small when $|k - j_0|$ is large. We combine this with the requirement of local connectivity, which implies that $W(j,k)$ is small when $|j - k|$ is large, and expand $W$ and $\mathbf{v}_\lambda$ to first-order in $|k - j_0|$ and $|j - k|$ respectively. With this approximation, we solve for $\mathbf{v}_\lambda$ across all nodes and find ('Methods' and mathematical appendix [*Supplementary file 1*], Section 2)

$$\mathbf{v}_\lambda(j) = e^{-\frac{(j-j_0)^2}{2\alpha(j_0,\omega)^2} + i\omega j}. \tag{2}$$

The eigenvector is a modulated Gaussian function, centered at $j_0$. The characteristic width is $\alpha$, such that a small $\alpha$ corresponds to a sharply localized eigenvector. Note that $j_0$ and $\omega$ depend on the particular timescale (or eigenvalue, $\lambda$) being considered and hence, in general, $\alpha^2$ will depend on the timescale under consideration. For $\mathbf{v}_\lambda$ to be localized, the real part of $\alpha^2$ must be positive when evaluated at the corresponding timescale. In this case, $\mathbf{v}_\lambda$ is consistent with our prior assumption, and we accept it as a meaningful solution.

Our theory gives the dependence of the eigenvector width on network parameters and on the corresponding timescale. In particular, $\alpha$ depends inversely on the degree of local heterogeneity in the network, so that greater heterogeneity leads to more tightly localized eigenvectors (see appendix [*Supplementary file 1*], Section 2). $\omega$ is a frequency term that allows $\mathbf{v}_\lambda$ to oscillate across nodes, as in *Equation 1*. As shown later, the method is general and a second-order expansion can be used when the first-order expansion breaks down. In that case the eigenvector shape is no longer Gaussian.

We now apply this theory to models of neural dynamics in the mammalian cerebral cortex. We use connectivity that decays exponentially with distance (*Markov et al., 2011*, *2014*; *Ercsey-Ravasz et al., 2013*) but our analysis applies to other forms of local connectivity.

## Localization in a network with a gradient of local connectivity

Our first model architecture is motivated by observations that as one progresses from sensory to prefrontal areas in the primate brain, neurons receive an increasing number of excitatory connections from their neighbors (*Wang, 2001*; *Elston, 2007*; *Wang, 2008*). We model a chain of nodes (i.e., neurons, networks of neurons or cortical areas) with connectivity that decays exponentially with distance. In addition, we introduce a gradient of excitatory self-couplings along the chain to account for the increase in local excitation.

The network is shown in *Figure 3A* and the coupling matrix $W$ is given by

$$W(j,k) = \begin{cases} \mu_0 + \Delta_r j & \text{for } j = k \text{ (self-coupling)} \\ \mu_f e^{-(j-k)/l_c} & \text{for } j > k \text{ (feedforward connections)}. \\ \mu_b e^{(j-k)/l_c} & \text{for } j < k \text{ (feedback connections)} \end{cases} \tag{3}$$

The self-coupling includes a leakage term ($\mu_0 < 0$) and a recurrent excitation term that increases along the chain with a slope $\Delta_r$. Nodes higher in the network thus have stronger self-coupling. Connection strengths have a decay length $l_c$. $\mu_f$ scales the overall strength of feedforward connections (i.e., connections from early to late nodes in the chain) while $\mu_b$ scales the strength of feedback connections. In general we set $\mu_f > \mu_b$.

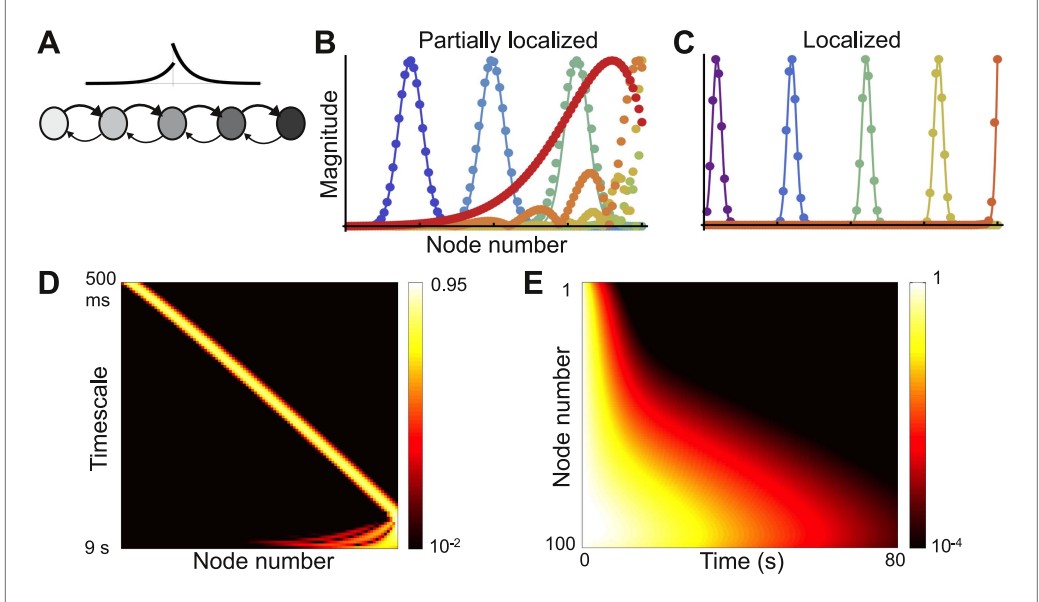

**Figure 3**. Localized eigenvectors in a network with a gradient of local connectivity. (**A**) The network is a chain of 100 nodes. Network topology is shown as a schematic with a subset of nodes and only nearest-neighbor connections. The plot above the chain shows the connectivity profile, highlighting the exponential decay and the asymmetry between feedforward and feedback connections. Self-coupling increases along the chain, as shown by the grayscale gradient. (**B**) Sample eigenvectors (filled circles) in a network with a weak gradient of self-coupling, so that localized and delocalized eigenvectors coexist. Localized eigenvectors are described by Gaussians, and predictions from *Equation 4* are shown as solid lines. Eigenvectors are normalized by maximum value. The network is described by *Equation 3*, with $\mu_0 = -1.9$, $\Delta_r = 0.0015$, $\mu_f = 0.2$, $\mu_b = 0.1$ and $l_c = 4$. (**C**) Sample eigenvectors (filled circles) along with predictions (solid lines) in a network with a strong gradient, so that all eigenvectors are localized. Network parameters are the same as **B**, except $\Delta_r = 0.01$. (**D**) Heat map of eigenvectors from network in (**C**) on logarithmic scale. Eigenvectors are along rows, arranged by increasing decay time. All are localized, and eigenvectors with longer timescales are localized further down in the chain. Edge effects cause the Gaussian shape to break down at the end of the chain, but eigenvectors are still localized at the boundary. (**E**) Dynamical response of the network in (**C**) to an input pulse. Nodes early in the chain show responses that decay away rapidly, while those further in the chain show more persistent responses.

The following figure supplements are available for figure 3:

**Figure supplement 1**. Co-existence of localized and delocalized eigenvectors in a network with a weak gradient of local connectivity.

If the gradient of self-coupling ($\Delta_r$) is strong enough, some of the eigenvectors of the network will be localized. As the gradient becomes steeper this region of localization expands. Our theory predicts which eigenvectors will be localized and how this region expands as the gradient becomes steeper (*Figure 3—figure supplement 1*).

By applying the theory sketched in the previous section (and developed in detail in the appendix [*Supplementary file 1*]), we find that the value of the eigenvector width for the localized eigenvectors ($\alpha$ in *Equation 2*) is equal to (see Section 3 of *Supplementary file 1*)

$$\alpha^2 = \frac{\mu_f - \mu_b}{2\Delta_r\left(1 + \cosh\left(\frac{1}{l_c}\right)\right)}. \tag{4}$$

This equation asserts that $\alpha^2$ is inversely proportional to the gradient of local connectivity, $\Delta_r$, so that a steeper gradient leads to sharper localization, and $\alpha^2$ increases with increasing connectivity decay length, $l_c$. Note that in this case the eigenvector width is independent of the location of the eigenvector (or the particular timescale).

In *Figure 3B*, we plot sample eigenvectors for a network with a weak gradient, where localized and delocalized eigenvectors coexist. We also plot the analytical prediction for the localized eigenvectors, which fits well with the numerical simulation results. For more details on this network see *Figure 3—figure supplement 1*. In *Figure 3C*, we plot sample eigenvectors for a network with a strong enough gradient that all eigenvectors are localized. As shown in *Figure 3D*, all the remaining eigenvectors of this network are localized. In *Figure 3E*, we plot the decay of this network's activity from a uniform initial condition; as predicted from the structure of the eigenvectors, decay time constants increase up the chain.

With a strong gradient of self-coupling, *Equation 4* holds for all eigenvectors except those at the end of the chain, where edge effects change the shape of the eigenvectors. These eigenvectors are still localized, at the boundary, but are no longer Gaussian and appear to be better described as modulated exponentials. *Equation 4* also predicts that eigenvectors become more localized as feed-forward and feedback connection strengths approach each other. This is counter-intuitive, since increasing feedback strength should couple nodes more tightly. Numerically, this prediction is confirmed only when $\mu_f - \mu_b$ is not close to 0. As seen in *Figure 4*, when $\mu_f - \mu_b$ is small, the eigenvector is no longer Gaussian and instead shows multiple peaks. Strengthening the feedback connections leads

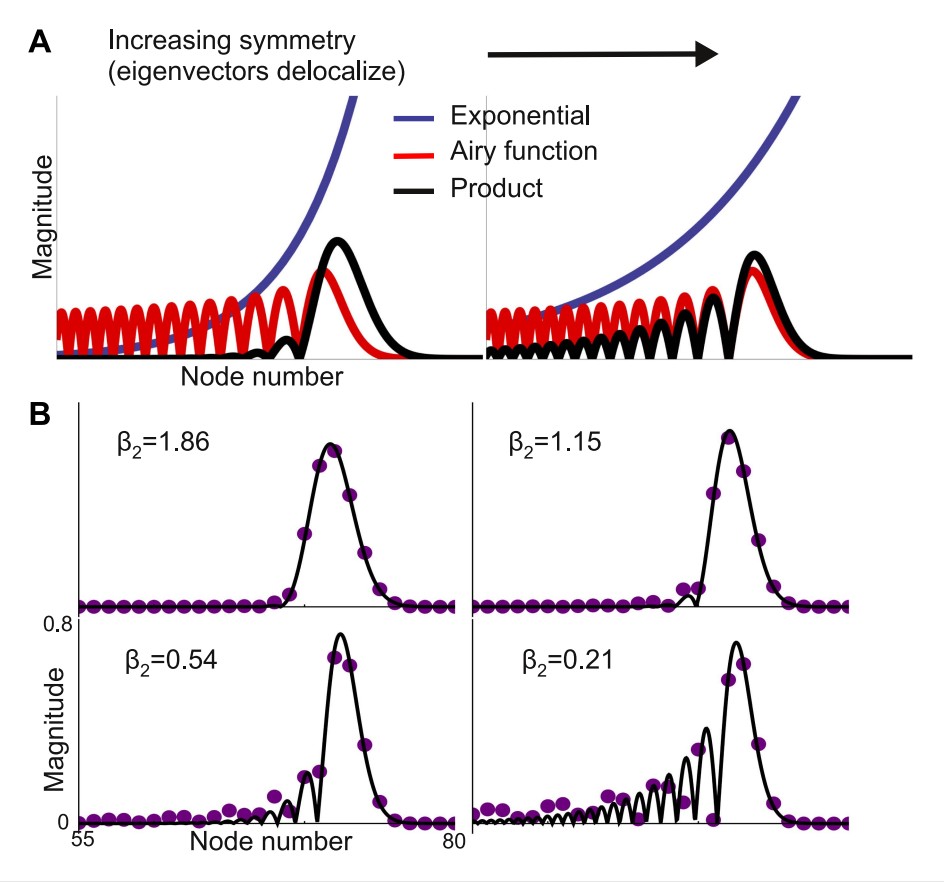

**Figure 4**. Second-order expansion for partially-delocalized eigenvectors. Same model with a gradient of local connectivity as in *Figure 3*. (**A**) Schematic of the predicted shape. Eigenvectors (black) are the product of an exponential (blue) and an Airy function (red). The constant in the exponential depends on the asymmetry between feedback ($\mu_b$) and feedforward ($\mu_f$) strengths. In the left panel, $\mu_f - \mu_b$ is large and the product is well described by a Gaussian. In the right panel, $\mu_f - \mu_b$ is small and the exponential is shallow enough that the product is somewhat delocalized. (**B**) Analytically predicted eigenvector shapes (solid lines) compared to numerical simulations (filled circles) for four values of $\mu_b$. For each value of $\mu_b$ one representative eigenvector is shown. As $\mu_b$ approaches $\mu_f$, eigenvectors start to delocalize but, as per *Equation 4*, the maximum peak is sharper. $\beta_2$ is the steepness of the exponential (*Equation 5*). The network is described by *Equation 3* with $\mu_0 = -1.9$, $\Delta_r = 0.01$, $\mu_f = 0.2$, and $l_c = 4$. $\mu_b = 0.125$, 0.15, 0.175, and 0.19.

to the emergence of ripples in the slower modes that modulate the activity of the earlier, faster nodes. While the first-order approximation of the shape of $v_\lambda$ breaks down in this regime, *Equation 4* is locally valid in that the largest peak sharpens with increasing symmetry, as seen in *Figure 4B*.

We extend our expansion to second-order in $v_\lambda$ (appendix [*Supplementary file 1*], Sections 5 & 6) to predict that the eigenvector is given by

$$\mathbf{v}_\lambda(j) = e^{\beta_2(j-j_0)} \mathrm{Ai}\left(\frac{\beta_1(j-j_0)+\beta_2^2}{\beta_1^{2/3}}\right) e^{i\omega j} \tag{5}$$

with

$$\beta_1 = \frac{\Delta_r \mathrm{csch}\left(\frac{1}{2l_c}\right)^4 \sinh\left(\frac{1}{l_c}\right)^3}{(\mu_f + \mu_b)} \quad \text{and} \quad \beta_2 = \frac{(\mu_f - \mu_b)\coth\left(\frac{1}{2l_c}\right)}{(\mu_f + \mu_b)} \tag{6}$$

where, Ai is the first Airy function (*Olver, 2010*). The eigenvector is the product of an exponential and an Airy function and this product is localized when the exponential is steep (*Figure 4A*). The steepness of the exponential depends on $\mu_f - \mu_b$. When this difference is small the exponential is shallow and the trailing edge of the product is poorly localized. *Figure 4B* shows that this functional form accurately predicts the results from numerical simulations, except when the eigenvector is almost completely delocalized.

These results reveal that an asymmetry in the strength of feedforward and feedback projections can play an important role in segregation of timescales in biological systems.

The second-order expansion demonstrates that the approach is general and can be extended as needed. While the first-order expansion in $v_\lambda$ generically gives rise to modulated Gaussians, the functional form of the eigenvectors from a second-order expansion depends on the connectivity (appendix [*Supplementary file 1*], Section 5) and, in general, the asymptotic decay is slower than that of a Gaussian.

## Localization in a network with a gradient of connectivity range

The previous architecture was a chain of nodes with identical inter-node connectivity but varying local connectivity. We now consider a contrasting architecture: a chain with no self-coupling but with a location-dependent bias in inter-node connectivity. We build this model motivated by the intuitive notion that nodes near the input end of a network send mostly feedforward projections, while nodes near the output send mostly feedback projections. The network architecture is shown in *Figure 5A*.

Connectivity decays exponentially, as in the previous example, but the decay length depends on position. Moving along the chain, feedforward decay length decreases while feedback decay length increases:

$$W(j,k) = \begin{cases} \mu_0 & \text{for } j = k \text{ (self-coupling)} \\ \mu_f e^{-(f_0 + f_1 k)(j-k)} & \text{for } j > k \text{ (feedforward connections)} \\ \mu_b e^{(b_0 - b_1 k)(j-k)} & \text{for } j < k \text{ (feedback connections)} \end{cases} \tag{7}$$

The parameters $f_0$, $f_1$, $b_0$, and $b_1$ control the location-dependence in decay length, $\mu_0$ is the leakage term, and $\mu_f$ and $\mu_b$ set the maximum strength of feedforward and feedback projections. We also add a small amount of randomness to the connection strengths.

As before we calculate the eigenvector width, $\alpha$. In this case, for a wide range of the parameters in *Equation 7*, $\alpha^2$ is positive and approximately constant for all eigenvectors. Therefore, all eigenvectors are localized and have approximately the same width (appendix [*Supplementary file 1*], Section 4). Four eigenvectors are plotted in *Figure 5B* along with theoretical predictions. *Figure 5C* shows all of the eigenvectors on a heat map and demonstrates that all are localized. The fastest and slowest timescales are localized to the earlier nodes while the intermediate timescales are localized towards the end of the chain. The earlier nodes thus show a combination of very fast and very slow time courses, whereas the later nodes display dynamics with an intermediate range of timescales. Such dynamics present a salient feature of networks with opposing gradients in their connectivity profile. In *Figure 5D*, we plot the decay of network activity from a uniform initial condition; note the contrast between nodes early and late in the chain.

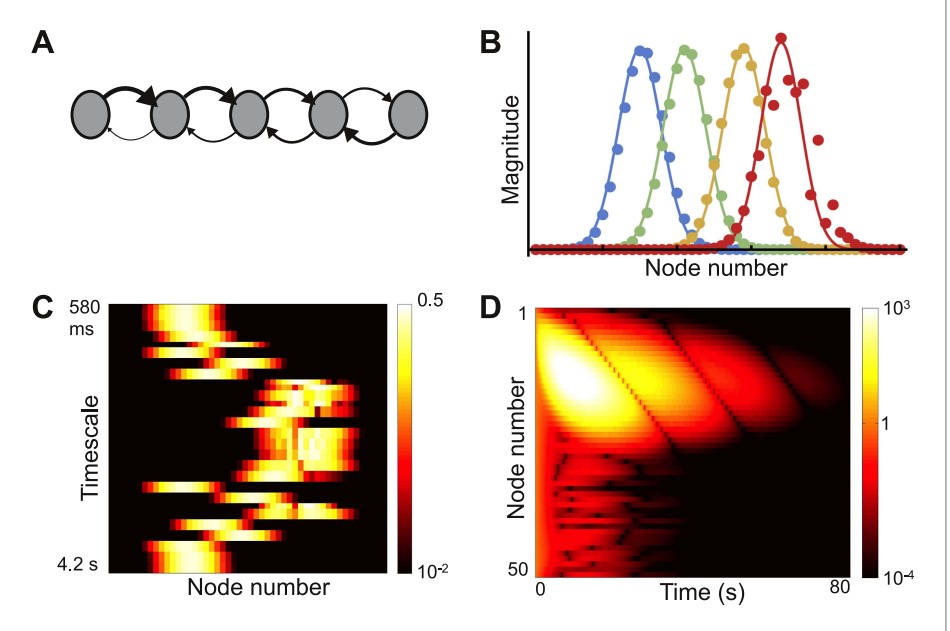

**Figure 5**. Localized eigenvectors in a network with a gradient of connectivity range. (**A**) The network consists of a chain of 50 identical nodes, shown here by a schematic. Spatial length of feedforward connections (from earlier to later nodes) decreases along the chain while the spatial length of feedback connections (from later to earlier nodes) increases along the chain. The network is described by *Equation 7*, with $\mu_0 = -1.05$, $\mu_f = 5$, $\mu_b = 0.5$, $f_0 = 0.2$, $f_1 = 0.12$, $b_0 = 6$, $b_1 = 0.11$. Normally-distributed randomness of standard deviation $\sigma = 10^{-5}$ is added to all connections. (**B**) Five sample eigenvectors, with numerical simulations (filled circles) well fitted by the analytical predictions (solid lines). Note the effect of added randomness on the rightmost eigenvector. (**C**) Heat map of eigenvectors on logarithmic scale. Rows correspond to eigenvectors, arranged by increasing decay time. All eigenvectors are localized, but timescales are not monotonically related to eigenvector position. (**D**) Dynamical response of the network to an input pulse. Long timescales are localized to nodes early in the network while nodes later in the network show intermediate timescales.

While the eigenvectors are all localized, different eigenvectors tend to cluster their centers near similar locations. Near those locations, nodes may participate in multiple eigenvectors, implying that time constants are not well segregated. This is a consequence of the architecture: nodes towards the edges of the chain project most strongly towards the center, so that small perturbations at either end of the chain are strongly propagated inward. The narrow spread of centers (the overlap of multiple eigenvectors) reduces the segregation of timescales that is one benefit of localization. We find that adding a small amount of randomness to the system spreads out the eigenvector centers without significantly changing the shape. This approach is more robust than fine-tuning parameters to maximally spread the centers, and seems reasonable in light of the heterogeneity intrinsic to biological systems (*Raser and O'Shea, 2005*; *Barbour et al., 2007*). Upon adding randomness, most eigenvectors remain Gaussian while a minority are localized but lose their Gaussian shape.

The significant overlap of the eigenvectors means that the eigenvectors are far from orthogonal to each other. Such matrices, called non-normal matrices, can show a number of interesting transient effects (*Trefethen and Embree, 2005*; *Goldman, 2009*; *Murphy and Miller, 2009*). In particular we note that the dynamics of our example network show significant initial growth before decaying, as visible in the scale of *Figure 5D*.

## Randomness and diversity

As observed in the last section, the heterogeneity intrinsic to biological systems can play a beneficial role in computation. Indeed, sufficient randomness in local node properties has been shown to give localized eigenvectors in models of physical systems with nearest-neighbor connectivity, and the transition from delocalized to localized eigenvectors has been suggested as a model of the transition from a conducting to an insulating medium (*Anderson, 1958*; *Abou-Chacra et al., 1973*; *Lee, 1985*).

A similar mechanism should apply in biological systems. We numerically explore eigenvector localization in a network with exponentially-decaying connectivity and randomly distributed self-couplings.

The network connection matrix is given by

$$W(j,k) = \begin{cases} \mu_0 + \mathcal{N}(0,\sigma^2) & \text{for } j = k \\ \mu_c e^{-|j-k|/l_c} & \text{for } j \neq k \end{cases},$$

(8)

where, $\mathcal{N}(0,\sigma^2)$ is drawn from a normal distribution with mean zero and variance $\sigma^2$.

As $\sigma^2$ increases, the network shows a transition to localization. This transition is increasingly sharp and occurs at lower values of $\sigma$ as the network gets larger. *Figure 6* shows a network with sufficient randomness for the eigenvectors to localize, with sample eigenvectors shown in *Figure 6B*. These show a variety of shapes and are no longer well described by Gaussians. Importantly, there is no longer a relationship between the location of an eigenvector and the timescale it corresponds to (*Figure 6C*). Thus while each timescale is localized, a variety of timescales are present in each region of the network, and each node will show a random mixture of timescales. This is in contrast to our previous examples, which have a spatially continuous distribution of time constants. The random distribution of time constants is also observed in the decay from a uniform initial conditions, as shown in *Figure 6D*.

## Discussion

Local connectivity is insufficient to create localized temporal patterns of activity in linear networks. A network with sharply localized but translationally invariant connectivity has delocalized eigenvectors. This implies that distant nodes in the network have similar temporal activity, since they share the timescales of their dynamics. Breaking the invariance can give rise to localized eigenvectors, and we study conditions that allow this. We develop a theory to predict the shapes of localized eigenvectors and our theory generalizes to describe eigenvectors that are only partially localized and show multiple peaks. A major

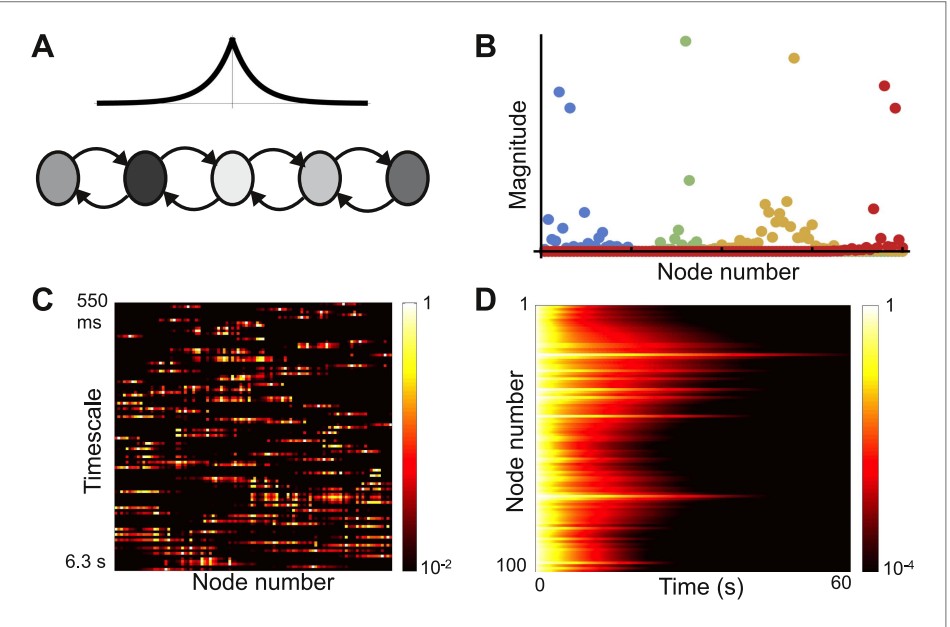

**Figure 6**. Localized eigenvectors in a network with random self-coupling. (**A**) The network consists of 100 nodes arranged in a chain. The plot above the chain shows the connectivity profile. Self-coupling is random, as indicated by the shading. The network is described by *Equation 8* with $\mu_0 = -1$, $\mu_c = 0.05$, $l_c = 4$, $\sigma = 0.33$. (**B**) Four eigenvectors are shown, localized to different parts of the network. Note the diversity of profiles. (**C**) Heat map of eigenvectors on logarithmic scale. Rows correspond to eigenvectors, arranged by increasing decay time. All eigenvectors are localized, though the extent of localization (the eigenvector width) varies; and there is no relationship between the timescale of an eigenvector and its spatial location in the network. (**D**) Dynamical response of the network to an input pulse. Note that the diversity of dynamical responses is more limited, and bears no relationship to spatial location.

finding of this study is the identification of two network architectures, with either a gradient of local connectivity or a gradient of long-distance connection length, that give rise to activity patterns with localized timescales.

Our approach to eigenvector localization is partly based on *Trefethen and Embree (2005)*; *Trefethen and Chapman (2004)*. The authors study perturbations of translationally invariant matrices and determine conditions under which eigenvectors are localized in the large-N limit. We additionally assume that the connectivity is local, since we are interested in matrices that describe connectivity of biological networks. This allows us to calculate explicit functional forms for the eigenvectors.

We stress that the temporal aspect of the network dynamics should not be confused with selectivity across space in a neural network. Even if temporal patterns are localized, a large proportion of network nodes may be active in response to a given input, albeit with distinct temporal dynamics. Conversely, even if temporal patterns are delocalized, nodes show similar dynamics yet may still be highly selective to different inputs and any stimulus could primarily activate only a small fraction of nodes in the network.

Our results are particularly relevant to understanding networks that need to perform computations requiring a wide spread of timescales. In general, input along a fast eigenvector decays exponentially faster than input along a slow eigenvector. To see this, consider a network with a fast and a slow timescale ($1/|\lambda_{fast}|$ and $1/|\lambda_{slow}|$), and having initial condition with components $a_{fast}$ and $a_{slow}$ along the fast and the slow eigenvectors respectively. As shown in *Equation 11*, the network activity will evolve as

$a_{fast}e^{-|\lambda_{fast}|t} + a_{slow}e^{-|\lambda_{slow}|t}$. For a node to show a significant fast timescale in the presence of a slower, more persistent timescale, the contribution of this slow timescale to the node must be small. This can happen in two ways, corresponding to the terms of *Equation 10*. If the input contributes little to the slower eigenvectors then their amplitudes will be small at all nodes. This requires fine-tuned input (exponentially smaller along the slow eigenvectors) and means that the slow timescales do not contribute significantly to any node. Alternately, as in the architectures we propose, the slow eigenvectors could be exponentially smaller at certain nodes; these nodes will then show fast timescales for most inputs, with a small slow component.

The architecture with a gradient of local connectivity (*Figure 3*) may explain some observations in the larval zebrafish oculomotor system (*Miri et al., 2011*). The authors observed a wide variation in the time constants of decay of firing activity across neurons, with more distant neurons showing a greater difference in time constants. They proposed a model characterized by a chain of nodes with linearly-decaying connectivity and a gradient of connection strengths, and found that different nodes in the model showed different timescales. Furthermore, the introduction of asymmetry to connectivity (with feedback connections weaker than feedforward connections) enhanced the diversity of timescales. This effect of asymmetry was also seen in an extension of the model to the macaque monkey oculomotor integrator (*Joshua et al., 2013*). Our work explains why such architectures allow for a diversity of timescales, and we predict that such gradients and asymmetry should be seen experimentally.

With a gradient of local connections, time constants increase monotonically along the network chain. By contrast, with a gradient of connectivity length (*Figure 5*), the relationship between timescales and eigenvector position is lawful but non-monotonic, as a consequence of the existence of two gradients (feedforward connectivity decreases while feedback increases along the chain). The small amount of randomness added to this system helps segregate the timescales across the network, while only mildly affecting the continuous dependence of eigenvector position on timescale. This suggests that randomness may contribute to a diversity of timescales.

The connection between structural randomness and localization is well known in physical systems (*Anderson, 1958*; *Abou-Chacra et al., 1973*; *Lee, 1985*). We applied this idea to a biological context (*Figure 6*), and showed that localization can indeed emerge from sufficiently random node properties. However, in this case nearby eigenvectors do not correspond to similar timescales. A given timescale is localized to a particular region of the network but a similar timescale could be localized at a distant region and, conversely, a much shorter or longer timescale could be localized in the same part of the network. Thus, the timescales shown by a particular node are a random sample of the timescales of the network.

Chemical gradients are common in biological systems, especially during development (*Wolpert, 2011*), and structural randomness and local heterogeneity are ubiquitous. We predict that biological systems could show localized activity patterns due to either of these mechanisms or a combination of the two. Furthermore, local randomness can enhance localization that emerges from gradients or long-range spatial fluctuations in local properties. We have focused on localization that yields a smooth

relationship between timescale and eigenvector position; such networks are well-placed to integrate information at different timescales. However, it seems plausible that biological networks have evolved to take advantage of randomness-induced localization, and it would be interesting to explore the computational implications of such localization. It could also be fruitful to explore localization from spatially correlated randomness.

An influential view of complexity is that a complex network combines segregation and integration: individual nodes and clusters of nodes show different behaviors and subserve different functions; these behaviors, however, emerge from network interactions and the computations depend on the flow of information through the network (*Tononi and Edelman, 1998*). The localized activity patterns we find are one way to construct such a network. Each node participates strongly in a few timescales and weakly in the others, but the shape and timescales of the activity patterns emerge from the network topology as a whole and information can flow from one node to another. Moreover, as shown in *Figure 7*, adding a small number of long-range strong links to local connectivity, as in small-world networks (*Watts and Strogatz, 1998*), causes a few eigenvectors to delocalize while leaving most localized. This is a possible mechanism to integrate computations while preserving segregated activity, and is an interesting direction for future research.

## Methods

We study the activity of a linear network of coupled units, which will be called 'nodes'. These represent neurons or populations of neurons. The activity of the $j$th node, $\phi_j(t)$, is determined by interactions with the other nodes in the network and by external inputs. It obeys the following equation:

$$\frac{d}{dt}\phi_j(t) = \sum_{k=1}^{N} W(j,k)\phi_k(t) + I_j(t),$$  (9)

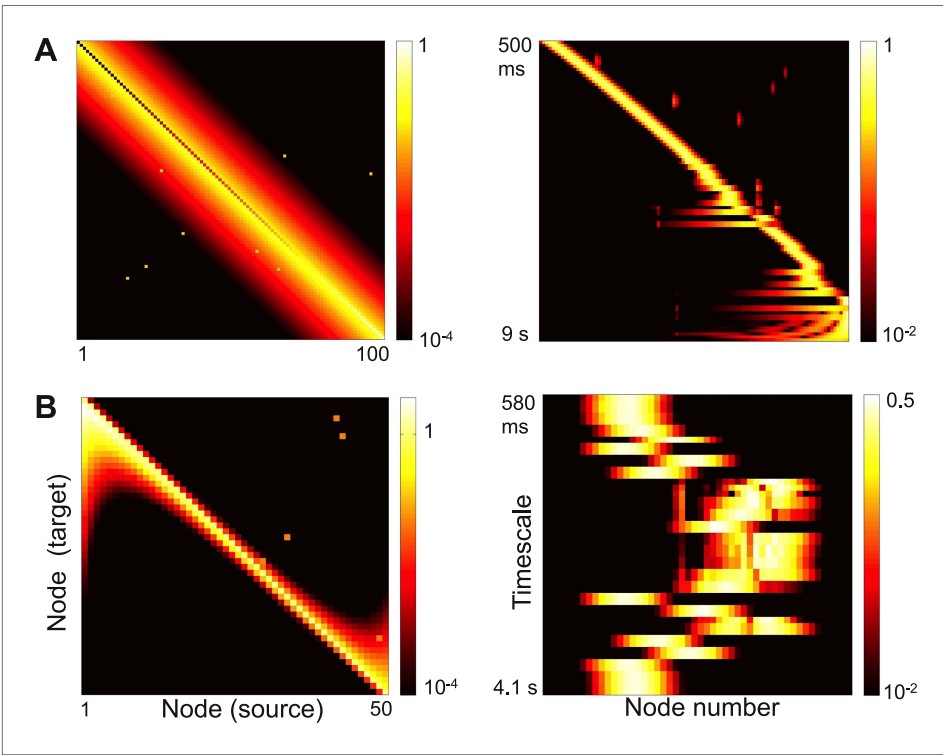

**Figure 7**. Strong long-range connections can delocalize a subset of eigenvectors. (**A**) Left panel: connectivity of the network in *Figure 3* with long-range connections of strength 0.05 added between 10% of the nodes. The gradient of self-coupling is shown along the diagonal on another scale, for clarity. Right panel: eigenvectors shown as in panel **C** of *Figure 3*. (**B**) Left panel: connectivity of the network in *Figure 5* with long-range connections of strength 0.05 added between 10% of the nodes. Right panel: eigenvectors shown as in panel **C** of *Figure 5*.

where $W(j,k)$ is the connection strength from node $k$ to node $j$ of the network and $I_j$ is the external input to the $j$th node. $W(j,j)$ is the self-coupling of the $j$th node and typically includes a leakage term. Note that the intrinsic timescale of node $j$ is absorbed into the matrix $W$.

By solving **Equation 9**, $\phi_j(t)$ can be expressed in terms of the eigenvectors of the connection matrix $W$, yielding

$$\phi_j(t) = \sum_\lambda A_\lambda(t) \mathbf{v}_\lambda(j) \tag{10}$$

(**Rugh, 1995**). Here, $\lambda$ indexes the eigenvalues of $W$, and $\mathbf{v}_\lambda(j)$ is the $j$th component of the eigenvector corresponding to $\lambda$. These are independent of the input. $A_\lambda(t)$ is the time-dependent amplitude of the eigenvector $\mathbf{v}_\lambda$ and depends on the input, which determines to what extent different eigenvectors are activated. If the real parts of the eigenvalues are negative then the network is stable and, in the absence of input, $A_\lambda(t)$ decays exponentially with a characteristic time of $1/\mathfrak{Re}(-\lambda)$.

$A_\lambda(t)$ consists of the sum of contributions from the initial condition and the input, so that **Equation 10** can be written as

$$\phi_j(t) = \sum_\lambda \left[ \tilde{a}_\lambda e^{\lambda t} + \int_0^t e^{\lambda(t-t')} \tilde{I}_\lambda(t') dt' \right] \mathbf{v}_\lambda(j). \tag{11}$$

$\tilde{a}_\lambda$ and $\tilde{I}_\lambda$ are the coefficients for the initial condition and the input, respectively, represented in the coordinate system of the eigenvectors. In a stable network, each node forgets its initial condition and simultaneously integrates input with the same set of time constants.

In this work, we examine different classes of the connection matrix $W$, with the constraint that connectivity is primarily local, and we identify conditions under which its eigenvectors are localized in the network in such a way that different nodes (or different parts of the network) exhibit disparate timescales.

## The functional form of localized eigenvectors from a first-order expansion

We rewrite the connectivity matrix in terms of a relative coordinate, $p = j-k$, as

$$W(j,k) = c(j, j-k). \tag{12}$$

Thus, $c(j,2) = W(j,j-2)$ indexes feedforward projections that span two nodes, and $c(5,p) = W(5,5-k)$ indexes projections to node 5. Note that in the translation-invariant case, $c(j,p)$ would be independent of $j$ (appendix [**Supplementary file 1**], Section 1), while the requirement of local connectivity means that $c(j,p)$ is small away from $p = 0$. For any fixed $j$, $c(j,p)$ is defined from $p = j - N$ to $p = j - 1$. We extend the definition of $c(j,p)$ to values outside this range by defining $c(j,p)$ to be periodic in $p$, with the period equal to the size of the network. This is purely a formal convenience to simplify the limits in certain sums and does not constrain the connectivity between the nodes of the network.

Consider the candidate eigenvector $\mathbf{v}_\lambda(j) = g_\lambda(j) e^{i\omega j}$. The dependence of $g_\lambda$ on $j$ allows the magnitude of the eigenvector to depend on position; setting this function equal to a constant returns us to the translation-independent case (see appendix [**Supplementary file 1**], Section 1). Moreover, note that $g_\lambda(j)$ depends on $\lambda$, meaning that eigenvectors corresponding to different eigenvalues (timescales) can have different shapes. For example, different eigenvectors can be localized to different degrees, and localized and delocalized eigenvectors can coexist (see **Figure 3—figure supplement 1** for an illustration). $\omega$ allows the eigenvector to oscillate across nodes; it varies between eigenvectors and so depends on $\lambda$.

Applying $W$ to $\mathbf{v}_\lambda$ yields

$$[W\mathbf{v}_\lambda](j) = \sum_{k=1}^N W(j,k) g_\lambda(k) e^{i\omega k} = \sum_{k=1}^N c(j, j-k) g_\lambda(k) e^{i\omega k} \tag{13}$$

$$= \left( \sum_{p=j-N}^{j-1} c(j,p) g_\lambda(j-p) e^{-i\omega p} \right) e^{i\omega j}, \tag{14}$$

here, the term in brackets is no longer independent of $j$.

So far we have made no use of the requirement of local connectivity and, given that $g_\lambda$ is an arbitrary function of position and can be different for different timescales, we have placed no constraints on the shape of the eigenvectors. By including an oscillatory term ($e^{i\omega j}$) in our ansatz, we ensure that $g_\lambda(j)$ is constant when connectivity is translation-invariant; this will simplify the analysis.

We now approximate both $c(j,p)$ and $g_\lambda(j-p)$ to first-order (i.e., linearly):

$$c(j,p) \approx c(j_0,p) + \frac{\partial c}{\partial j}\Big|_{j_0,p}(j-j_0)$$

$$g_\lambda(j-p) \approx g_\lambda(j) - g'_\lambda(j)p, \tag{15}$$

where, $j_0$ is a putative center of the eigenvector.

Substituting *Equation 15* into *Equation 14* we get

$$[W\mathbf{v}_\lambda](j) = \left(\sum_{p=j-N}^{j-1}\left[c(j_0,p) + \frac{\partial c}{\partial j}\Big|_{j_0,p}(j-j_0)\right]\left[g_\lambda(j) - g'_\lambda(j)p\right]e^{-i\omega p}\right)e^{i\omega j} \tag{16}$$

We expect these approximations to be valid only locally. However, if connectivity is local then the major contribution to the sum comes from small values of $p$. For large values of $p$, $g_\lambda(j-p)$ is multiplied by connectivity strengths close to 0 and so we only need to approximate $g_\lambda$ for $p$ close to 0. Similarly, in approximating $c(j,p)$ around $j=j_0$, we expect our approximation to be good in the vicinity of $j=j_0$. However, if our eigenvector is indeed localized around $j_0$, then $g_\lambda(k)$ is small when $|k-j_0|$ is large. For small $p$, large values of $|k-j_0|$ approximately correspond to large values of $|j-j_0|$, and so $c(j,p)$ makes a contribution to the sum only when $j \approx j_0$.

The zeroth-order term in *Equation 16* is

$$\left(\sum_{p=j-N}^{j-1}c(j_0,p)e^{-i\omega p}\right)g_\lambda(j)e^{i\omega j} = \lambda(j_0,\omega)\mathbf{v}_\lambda(j)$$

The function in parentheses is periodic in $p$ with period $N$ (recall that $c(j,p)$ was extended to be periodic in $p$). Thus to zeroth-order $\mathbf{v}_\lambda$ is an eigenvector with eigenvalue

$$\lambda(j_0,\omega) = \sum_{p=1}^{N}c(j_0,p)e^{-i\omega p}. \tag{17}$$

For $\lambda$ to be an exact eigenvalue in *Equation 16*, the higher-order terms should vanish. By setting the first-order term in this equation to 0, we obtain a differential equation for $g_\lambda(j)$:

$$-\alpha(j_0,\omega)^2 g'_\lambda(j) = (j-j_0)g_\lambda(j) \tag{18}$$

where,

$$\alpha(j_0,\omega)^2 = -\frac{\sum_p pc(j_0,p)e^{-i\omega p}}{\sum_p \frac{\partial c}{\partial j}\Big|_{j_0,p}e^{-i\omega p}}. \tag{19}$$

Thus $\alpha^2$ is a ratio of discrete Fourier transforms at the frequency $\omega$. Note that the denominator is a weighted measure of network heterogeneity at the location $j_0$. Also note that $\alpha^2$ can be written in terms of $\lambda$ as (compare the twist condition of *Trefethen and Embree, 2005*):

$$\alpha^2(j_0,\omega) = -i\frac{\frac{\partial \lambda}{\partial \omega}}{\frac{\partial \lambda}{\partial j_0}}. \tag{20}$$

Solving for $g\lambda$ in *Equation 18* yields

$$g_\lambda(j) = C_1 e^{-\frac{(j-j_0)^2}{2\alpha(j_0,\omega)^2}},$$

where, $C_1$ is a constant. Thus, to first-order, the eigenvector is given by the modulated Gaussian function

$$\mathbf{v}_\lambda(j) = e^{-\frac{(j-j_0)^2}{2\alpha(j_0,\omega)^2} + i\omega j}.$$ (21)

In general, $\alpha$ can be complex. In order for $\mathbf{v}_\lambda$ to be localized, $\mathfrak{Re}(\alpha^2)$ must be positive for the corresponding values of $j_0$ and $\omega$, and we only accept an eigenvector as a valid solution if this is the case. Thus the approach is self-consistent: we assumed that there existed a localized eigenvector, combined this with the requirement of local connectivity to solve for its putative shape, and then restricted ourselves to solutions that did indeed conform to our initial assumption.

For an expanded version of this analysis along with further discussion of what the analysis provides, see the appendix (*Supplementary file 1*), Section 2.

## Additional information

### Funding

| Funder | Grant reference number | Author |
| --- | --- | --- |
| Office of Naval Research | N00014-13-1-0297 | Xiao-Jing Wang |
| John Simon Guggenheim Memorial Foundation Fellowship | | Xiao-Jing Wang |

The funders had no role in study design, data collection and interpretation, or the decision to submit the work for publication.

### Author contributions

RC, AB, Conception and design, Acquisition of data, Analysis and interpretation of data, Drafting or revising the article; X-JW, Conception and design, Analysis and interpretation of data, Drafting or revising the article

## Additional files

### Supplementary files

• Supplementary file 1. Mathematical appendix.

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
