## [Decision Letter]

Thank you for sending your work entitled “A diversity of localized timescales in network activity” for consideration at *eLife*. Your article has been favorably evaluated by a Senior editor and 2 reviewers, one of whom is a member of our Board of Reviewing Editors.

The Reviewing editor and the other reviewers discussed their comments before we reached this decision, and the Reviewing editor has assembled the following comments to help you prepare a revised submission.

This is a very interesting article from a theoretical perspective. It shows how a combination of non-Hermiticity and broken translation invariance can lead generically to surprisingly localized eigenfunctions. Biological implications of this result are that neuronal networks in the brain could have localized modes of activation characterized by different time scales.

The main issue that should be addressed in the revision is that, while on one hand, the analytical method for estimation of eigenvectors is the major contribution, the method is presented in an incomprehensible manner. As a result, one cannot appreciate why is it advantageous to simply diagonalize the connectivity matrix numerically (this issue is not discussed by the authors). Here is the list of points that have to be clarified.

1) [Disp-formula equ8] - it has to be solved for j_0_ and w to find the shape of the corresponding eigenvector. How do we know the eigenvalues of the matrix? This is never explained. Moreover, in the first example considered, of [Disp-formula equ14], the authors simply say that they ‘match the eigenvalues to j_0_ and w, to find that w=pi’. Is there an analytical expression for the eigenvalues of the matrix of [Disp-formula equ14]? If yes, the authors should provide it. It would make this example very special though. What if there is no such expression, would they have to diagonalize the matrix numerically? This would also provide the eigenvectors, so the whole procedure would seem to be redundant.

2) In the next example, of [Disp-formula equ21], apparently there is no analytical expression for the eigenvalues, and the final solution for the width of the eigenvactor, [Disp-formula equ25] still depends on j_0_ and w. The authors don’t explain how they find those.

3) In the presentation of the second-order expansion approach, the authors seem to ignore that both first-order and second-order corrections to eigenvalues have to vanish. They only consider the second order. Why does it makes sense to ignore the first-order correction?

---

## [Author Response]

*1)*
[Disp-formula equ8]
*- it has to be solved for j_0_ and w to find the shape of the corresponding eigenvector. How do we know the eigenvalues of the matrix? This is never explained. Moreover, in the first example considered, of*
[Disp-formula equ14]*, the authors simply say that they ‘match the eigenvalues to j_0_ and w, to find that w=pi’. Is there an analytical expression for the eigenvalues of the matrix of*
[Disp-formula equ14]*? If yes, the authors should provide it. It would make this example very special though. What if there is no such expression, would they have to diagonalize the matrix numerically? This would also provide the eigenvectors, so the whole procedure would seem to be redundant.* And

*2) In the next example, of*
[Disp-formula equ21]*, apparently there is no analytical expression for the eigenvalues, and the final solution for the width of the eigenvactor,*
[Disp-formula equ25]
*still depends on j_0_ and w. The authors don’t explain how they find those*.

We thank the reviewers for this point—the discussion of the relationship to the eigenvalues was unclear and has now been clarified. We start by stressing that the benefit of our approach is not primarily computational. The reviewers are right that, in general, our method doesn't provide a way to analytically compute the eigenvalues and, given that we consider a large class of matrices, it would be surprising if we could. Instead, our approach yields theoretical insight into the conditions that allow for eigenvector localization and how the shape of localized eigenvectors depend on network parameters.

In Supplementary file 1 (mathematical appendix), we have added an extensive discussion of what the theory yields and why it is useful. We have also clarified this in the main text and, to avoid any confusion, highlighted that our theory does not in general predict the eigenvalues of an arbitrary matrix with local connectivity. Finally, we have added a figure (Figure 3—figure supplement 1) that demonstrates how the theory picks out a region of the complex plane within which localized eigenvectors lie. We summarize these points below.

Given a network specification (i.e., connectivity profile), our analysis reveals the functional form of the eigenvector and, to first order, localized eigenvectors are Gaussians. However, the parameters of this functional form (in this case the center and width) depend on the particular connectivity profile (known) and on the eigenvalues, which are unknown. In general these eigenvalues must be separately calculated. Given a particular eigenvalue the theory tells us whether the corresponding eigenvector is localized. In this case it also yields the shape along with an analytic formula for the dependence of the shape on network parameters; this formula can be used to understand how changing network parameters promotes or hinders localization.

Our theory also identifies a region of the complex plane within which the eigenvalues lie, and tells us which of these putative timescales will correspond to localized eigenvectors. It tells us which nodes can host localized eigenvectors and how changing the parameters of the network changes the region of localized timescales. It also provides qualitative insight into factors (like translation-dependence and asymmetry) that promote localization.

In certain cases we can draw general conclusions about the shapes of all localized eigenvectors without computing the eigenvalues. For example, in the network of Figure 3, the eigenvector width is the same for all localized eigenvectors. This is a special example but not an unnatural one; a gradient of local properties is among the simplest deterministic ways to break translation-invariance. We also note that our theory allows us to translate constraints on the eigenvalue spectrum of the network (for example, low-rank or sharply-decaying connectivity, real eigenvalues, etc) into constraints on the shape of eigenvectors.

We elaborate on these points in Section 2 of the mathematical appendix.

*3) In the presentation of the second-order expansion approach, the authors seem to ignore that both first-order and second-order corrections to eigenvalues have to vanish. They only consider the second order. Why does it makes sense to ignore the first-order correction*?

In all of our expansions, we require that the sum of the higher order terms vanishes. For the second-order expansion, this means that the sum of the first and second-order terms should vanish. Note that this does not necessarily mean that the terms vanish separately.